# Influence of Equimolar Doses of Beetroot Juice and Sodium Nitrate on Time Trial Performance in Handcycling

**DOI:** 10.3390/nu11071642

**Published:** 2019-07-18

**Authors:** Joelle Leonie Flueck, Alessandro Gallo, Nynke Moelijker, Nikolay Bogdanov, Anna Bogdanova, Claudio Perret

**Affiliations:** 1Institute of Sports Medicine, Swiss Paraplegic Centre Nottwil, CH-6207 Nottwil, Switzerland; 2Department of Health and Technology, ETH Zurich (Swiss Federal Institute of Technology Zurich), CH-8003 Zurich, Switzerland; 3Institute of Veterinary Physiology, Vetsuisse Faculty and the Zurich Center for Integrative Human Physiology (ZIHP), University of Zurich, CH-8057 Zurich, Switzerland

**Keywords:** Paralympic, sports nutrition, supplementation

## Abstract

This study aimed to investigate the influence of a single dose of either beetroot juice (BR) or sodium nitrate (NIT) on performance in a 10 km handcycling time trial (TT) in able-bodied individuals and paracyclists. In total, 14 able-bodied individuals [mean ± SD; age: 28 ± 7 years, height: 183 ± 5 cm, body mass (BM): 82 ± 9 kg, peak oxygen consumption (VO_2peak_): 33.9 ± 4.2 mL/min/kg] and eight paracyclists (age: 40 ± 11 years, height: 176 ± 9cm, BM: 65 ± 9 kg, VO_2peak_: 38.6 ± 10.5 mL/min/kg) participated in the study. All participants had to perform three TT on different days, receiving either 6 mmol nitrate as BR or NIT or water as a placebo. Time-to-complete the TT, power output (PO), as well as oxygen uptake (VO_2_) were measured. No significant differences in time-to-complete the TT were found between the three interventions in able-bodied individuals (*p* = 0.80) or in paracyclists (*p* = 0.61). Furthermore, VO_2_ was not significantly changed after the ingestion of BR or NIT in either group (*p* < 0.05). The PO to VO_2_ ratio was significantly higher in some kilometers of the TT in able-bodied individuals (*p* < 0.05). The ingestion of BR or NIT did not increase handcycling performance in able-bodied individuals or in paracyclists.

## 1. Introduction

Sports nutrition and performance enhancing supplements are prominently discussed topics in today’s sports science. Currently, dietary nitrate is one of the latest trends in sports nutrition [1]. The main sources of dietary nitrate are green leafy and root vegetables, including beetroot, spinach, and rocket [2]. After ingestion, nitrate (NO_3_^−^) is reduced to nitrite (NO_2_^−^) and then to nitric oxide (NO), leading to a temporary increase in plasma and muscle NO [3]. Production of NO from nitrate and nitrite is enhanced under hypoxic and acidic conditions [4]. The role of NO as a signaling molecule is that it interacts with multiple targets (e.g., heme and thiol groups of proteins), triggering numerous downstream events, among which stimulation of protein kinase G and control over mitochondrial function are just a few [5]. At the organism level, NO controls heart and muscle function [6,7], vascular tone [7,8], O_2_ transport by red blood cells, and other processes important for sports performance [4]. It has been proven that dietary nitrate supplementation reduces blood pressure in healthy individuals as well as in patients suffering from hypertension, diabetes, or ischemia [9,10,11]. Besides its beneficial therapeutic application, dietary nitrate supplementation, either as nitrate-rich beetroot juice (BR) or as sodium nitrate (NIT), has become a popular sport supplement based on its ergogenic effects (for review, see [12,13,14]).

Several studies have investigated the use of dietary nitrate for athletes in different disciplines, predominantly focusing on lower-body or whole-body exercises, including cycling [15,16] and running [17,18]. It has been demonstrated that dietary nitrate supplementation can improve both exercise capacity (e.g., time-to-exhaustion) [19,20] as well as exercise performance (e.g., time-to-completion) [16,21]. In addition, nitrate supplementation has been shown to reduce the oxygen cost (VO_2_) of exercise [19,22]. It is assumed that these observed effects of nitrate supplementation are a consequence of an improved mitochondrial efficiency (reduced oxygen cost of ATP production) [23] and/or due to enhanced muscle contractile functions [24,25]. Complementary nitrate supplementation might promote vasorelaxation and elevate skeletal muscle oxygen delivery during exercise, as shown in rodents [26]. However, it seems unclear whether vasodilatation might improve performance [27,28]. Noteworthy targeted effects of nitrate supplementation on fast-twitch muscle fibers have been demonstrated previously [29,30].

Fat-twitch muscle fibers are more abundant in the upper body compared to the lower body muscles [31,32], and they exhibit different cardiovascular regulation mechanisms during exercise [33]. Furthermore, previous investigations reported that the cardiovascular strain in arm cranking was much higher compared to leg pedaling [34], and that arm cranking relies more on glycolytic metabolism [35]. All these physiological features of the upper body suggest that it should be sensitive to nitrate supplementation. However, the effects of dietary nitrate on upper body exercise performance have never been investigated in detail.

To date, only few studies have examined the effect of dietary nitrate supplementation on upper body exercise performance such as kayaking [36,37] or rowing [38,39]. These studies reported inconsistent results. Some investigations reported that BR supplementation improved repeated sprint and 2 km rowing performance in well trained athletes [38,39]. Furthermore, Peeling et al. [37] demonstrated that 500 m kayaking performance was significantly improved after the intake of BR. On the other hand, Muggeridge et al. [36] could not observe any ergogenic effects of BR supplementation on repeated sprint or 1 km time trial (TT) performance in trained kayakers.

To our knowledge, the effects of dietary nitrate supplementation on handcycling performance were never examined. Therefore, we aimed to investigate whether a single dose of 6 mmol nitrate as BR or NIT supplement improved 10 km handcycling TT performance in recreationally active individuals compared to paracycling athletes. Furthermore, we aimed to assess the effects of dietary nitrate supplementation on VO_2_ during the TT. We hypothesized that, compared to placebo (PLA), a single intake of dietary nitrate would either (1) improve 10 km TT performance with similar VO_2_ or (2) reduce VO_2_ (whereas TT performance would remain unchanged).

## 2. Materials and Methods

### 2.1. Participants

Upper body trained able-bodied individuals as well as paracyclists with a spinal cord injury were recruited for this study. Athletes with a minimal training duration of 3 h per week and 3 sessions per week were admitted into the study. All participants were non-smokers. Participants using medication that might influence performance were excluded from the study. Furthermore, participants with nitrate hypersensitivity, diabetes, cardiovascular disease, pulmonary disease, or any other disease were not allowed to participate. Participants were aware of the study aim and the effects of nitrate. Before the start of the study, all participants provided written informed consent. The study was approved by the local ethical committee (EC-No. 2015-209, SNCTP-No. NCT02454049, Ethikkomission Nordwest- und Zentralschweiz, EKNZ, Basel, Switzerland) and met all ethical standards of the institutional and national research committee and the 1964 Helsinki Declaration [40].

### 2.2. Study Design

A randomized, placebo-controlled, single-blind, and cross-over study was performed at the Institute of Sports Medicine Nottwil, Switzerland. Each participant visited the exercise lab on five separate occasions. On the first visit, medical history was checked to fulfill all inclusion criteria, and participants were asked to carefully read and sign the written informed consent. On the second visit, a screening questionnaire was completed, and a subsequent incremental exercise test was performed to determine peak oxygen consumption (VO_2peak_) and maximal power output (PO_max_). Shortly after, a familiarization TT was performed. On the following three occasions, the participants performed a 10 km handcycling TT after the ingestion of BR, NIT, or PLA. Diet, sleep, training load, and acute sickness were recorded by a questionnaire before each trial to ensure equal testing conditions. All experimental trials were conducted within a maximum of six weeks, and TTs were at minimum 48 h apart for recovery purposes.

For the three experimental trials, participants arrived at the laboratory at approximately the same time of day (± 2 h). After a 5 min resting period and the completion of the questionnaire, resting blood pressure and heart rate were measured in triplets in a sitting position at the wrist using an automated cuff (Boso medistars, Bosch + Sohn GmbH, Jungingen, Germany). The forearm was placed at heart level. Subsequently, participants received a non-transparent flask containing the supplement, which they ingested immediately. Afterwards, a 3 h resting period followed, where participants could leave the laboratory but were asked to refrain from any strenuous physical activity. During this time, participants were allowed to drink water ad libitum but were not allowed to eat. Following the 3 h resting period, blood pressure and heart rate were measured again. Thereafter, participants started with the 10 km TT.

### 2.3. Physical Activity and Dietary Standardization

Participants were asked to maintain their regular training schedule as consistent as possible over the time course of the study and were instructed to avoid high-intensity training the last two days before each experimental trial. They were allowed to maintain their habitual diet but were requested to abstain from nitrate-enriched foods (e.g., beetroot, spinach, rocket, lettuce, etc.) the last two days before each test. Furthermore, participants were requested to abstain from caffeine and alcohol the last 12 h before each test and were advised to eat a standardized meal 1 h before the arrival at the laboratory. They were instructed to maintain their sleep constant at a minimum of 7 h. To prevent any modification in the degradation process of nitrate by commensal bacteria in the oral cavity, participants were instructed to abstain from anti-bacterial mouthwash on all testing days [41].

### 2.4. Supplementation

Supplements were ingested in a fluid form. The supplements were either BR (6 mmol nitrate), water with sodium nitrate (NIT) (6 mmol nitrate), or plain water (PLA). The BR was a special production by an external company and was delivered in the form of a standardized shot (Biotta AG, Tägerwilen, Switzerland). For the preparation of the 6 mmol NIT supplements, the amount of 510 mg of sodium nitrate (Pure sodium nitrate, POCH S.A., supplier: Pharmaserv LTD, Stansstad, Switzerland) was weighted and dissolved in 85 mL of plain water. All supplements were bottled in 85 mL non-transparent flasks. Shortly after each exercise test, participants were asked about the supplement they received. Optically, they could not distinguish between NIT and PLA but could set PLA/NIT apart from BR because of taste and color. Only 60% of the athletes chose the right supplement after the NIT and PLA test. All athletes were able to distinguish BR apart from the other supplements.

### 2.5. Exercise Testing

#### 2.5.1. Maximal Oxygen Uptake and Familiarization

Participants performed an incremental exercise ramp test on the handcycle ergometer Cyclus 2 (Avantronic, Leipzig, Germany) to detect VO_2peak_ and PO_max_. After a baseline measurement of 1 min at rest, a standardized warm-up of 2 min cycling at 20 W was performed. Subsequently, the test started at 20 W. The workload increased continuously by 10 W per minute. Participants cycled until exhaustion and were verbally encouraged. Participants cycled at a self-selected pedal rate (60–100 rpm) and were instructed to keep this frequency constant during the test. Pedal rates < 50 rpm were regarded as test termination. During the test, heart rate was monitored (S610i, Polar Electro Oy, Kempele, Finland). These data were analyzed with the Polar Pro Trainer 5 software (Polar Electro Oy, Kempele, Finland). The breath-by-breath pulmonary gas exchange data were continuously measured during the test and averaged over 15 s periods (Oxycon Pro, Jaeger GmbH, Würzburg, Germany). VO_2peak_ was determined as the highest 15 s VO_2_ value during the test. Ergometer configurations were recorded and maintained equally for all following exercise tests. After a 20 to 30 min break, participants performed a 10 km familiarization TT identical to the following experimental TT.

#### 2.5.2. Time Trial

Three hours after the ingestion of the supplement, participants performed a simulated 10 km TT (0.5% incline) on the Cyclus 2 with a provided handbike (able-bodied individuals) or with their own handbike (paracyclists). All TTs were performed with the same fixed gearing. Participants performed a 5 min warm-up at self-selected pedal rate and with the fixed gearing. After a 1 min measurement at rest for baseline parameters, the 10 km TT began. After the warm-up and at the end of the TT, a capillary blood sample was taken from the earlobe and analyzed for blood lactate concentrations ([Lac]) using an enzymatic amperometric chip sensor system (Biosen C-Line Clinic, EKF diagnostic GmbH, Cardiff, UK), and rating of perceived exertion (RPE) was recorded using a Borg scale ranging from 6–20 [42]. During the test, a computer screen was placed in front of participants, which displayed the distance they had already cycled. No information was given about completion time, and participants received no feedback on performance during or after the TT. Furthermore, no verbal encouragement was provided during the TT. Heart rate was monitored during the warm-up and the TT. Respiratory gas exchange parameters were measured continuously during the TT using the above mentioned metabolic cart. Shortly before each measurement, the metabolic cart was calibrated by automatic volume calibration and by gas calibration using a standardized gas containing 16% O_2_ and 5% CO_2_ (Pangas, Dagmarsellen, Switzerland).

### 2.6. Plasma Nitrate and Nitrite Measurements

Venous blood samples were drawn into lithium-heparin tubes (2.7 mL Monovette Lithium Heparin; Sarstedt, Sevelen, Switzerland). All samples were immediately centrifuged at 3000 r·min^−1^ (1549 g) and 4 °C for 10 min within 1 min of collection. Blood plasma was harvested and stored at −80 °C in 1.5 mL tubes (Eppendorf, Wesseling–Berzdorf, Germany). Plasma nitrate ([NO_3_^−^]) and nitrite ([NO_2_^−^]) concentrations were assessed by chemiluminescence NO analyzer (CLD 88 sp NO Analyzer, ECO Medics AG, Duernten, Switzerland). [NO_2_^−^] was reduced to NO in Brown’s solution (0.57 g 99.99% iodine; 1.62 g KI, 15 mL bi-distilled water, 200 mL acetic acid), and subsequent photons produced as NO interacted with O_3_ were detected. Detection of [NO_3_^−^] included one more step in which [NO_3_^−^] was quantitatively reduced to [NO_2_^−^] using World Precisions Instrument’s Nitralyzer kit (World Precision Instruments, Sarasota, FL, USA). Reduction of [NO_3_^−^] to [NO_2_^−^] was performed on copper-coated cadmium beads. Thereafter, total NO metabolites level (NO_x_) = ([NO_2_^−^] + [NO_3_^−^]) was measured, and [NO_3_^−^] concentrations were calculated as a difference between [NO_2_^−^] levels before and after the reduction step. Calibration solutions of sodium nitrite (NaNO_2_^−^) (50, 100, 150, and 200 nmol·L^−1^) and sodium nitrate (NaNO_3_^−^) (20, 40, 60, 80, and 100 µmol·L^−1^) were prepared in bi-distilled water to produce calibration plots.

### 2.7. Data Analysis and Statistics

The statistical analysis was performed using the IBM SPSS Statistics software for Windows (Version 24, IBM Corp., Armonk, NY, USA). Statistical significance was set at an α-level of 0.05. The analysis with the Q-Q plot and the Shapiro-Wilk test showed that data for the able-bodied individuals were normally distributed, and data from paracycling athletes were not normally distributed. The data were presented as mean ± standard deviation (SD). The data of the able-bodied participants were analyzed using parametric testing. To analyze changes in blood pressure and heart rate from pre- to post-ingestion between NIT, BR, and PLA, a one-way repeated-measures ANOVA was performed. Differences in completion time, mean PO, mean VO_2_, mean PO/VO_2_ ratio, and other cardiorespiratory parameters were assessed using a one-way repeated-measures ANOVA. Pairwise t-tests were used for post hoc analysis, and Bonferroni corrections were applied for multiple testing. The PO and PO/VO_2_ ratio data were graphed for the 10 km distance and analyzed using a two-way (supplement x distance) repeated-measures ANOVA with Bonferroni correction. In case of violation of sphericity, Greenhouse–Geisser correction was applied. The effect of order on TT performance was analyzed using a one-way repeated-measures ANOVA and pairwise t-tests with Bonferroni correction.

The data from paracyclists were analyzed using non-parametric testing. The Friedman test was applied to detect differences between the three groups for all parameters. If significant differences occurred, the Wilcoxon post-hoc test was applied with Bonferroni correction. For comparison between able-bodied individuals and paracyclists, the Mann-Whitney-U test was used to detect any differences.

## 3. Results

Fourteen healthy, recreationally active men [mean ± SD; age: 28 ± 7 years, height: 183 ± 5 cm, body mass (BM): 82 ± 9 kg, VO_2peak_: 33.9 ± 4.2 mL/min/kg, PO_max_: 152 ± 20 W] who were used to upper body exercises volunteered to participate in this study. They trained 4 ± 2 times per week with a total weekly duration of 6 ± 3 h. Additionally, eight paracyclists from the national team (data presented in Table 1) participated in this study. They trained 11 ± 4 h and 7 ± 2 times per week.

### 3.1. Blood Pressure and Heart Rate at Rest

No significant differences (*p* > 0.05) in systolic and diastolic blood pressure pre and post ingestion of the supplements between the interventions were found in able-bodied participants or paracyclists. Neither in able-bodied individuals nor in paracyclists did the change in blood pressure from pre to post ingestion significantly differ (*p* > 0.05) in all three interventions. No significant difference in systolic and diastolic blood pressure was found between able-bodied individuals and paracyclists (*p* > 0.05). No significant differences between the three interventions in able-bodied individuals and in paracyclists were observed for the change of heart rate from pre to post ingestion of the supplement (*p* < 0.05). No significant difference between able-bodied individuals and paracyclists was found (*p* < 0.05). Data are presented in Table 2.

### 3.2. Plasma Nitrate and Nitrite Concentrations Pre and Post Supplement Ingestion

Plasma [NO_2_^−^] and [NO_3_^−^] concentrations before and after the ingestion of NIT, BR, or PLA are illustrated in Table 3. No significant difference was found in plasma [NO_2_^−^] and [NO_3_^−^] pre supplement ingestion in able-bodied individuals (*p* = 0.99) or in paracyclists (*p* = 0.19). A significantly higher [NO_2_^−^] and [NO_3_^−^] concentration was found in able-bodied individuals (NO_2_^−^: *p* = 0.011; NO_3_^−^: *p* = 0.001) and in paracyclists (NO_2_^−^: *p* = 0.002; NO_3_^−^: *p* = 0.001) in the BR intervention compared to PLA. No significant difference was found post ingestion between plasma [NO_3_^−^] and [NO_2_^−^] concentration in BR and NIT in either group (*p* > 0.05). Able-bodied individuals showed significantly higher [NO_2_^−^] concentrations after BR consumption compared to paracyclists (*p* < 0.05). Paracyclists showed significantly higher [NO_3_^−^] concentrations after BR and NIT consumption compared to able-bodied individuals (*p* < 0.05).

### 3.3. Performance Parameters during Warm-up and TT

The warm-up parameters were not significantly different in the three intervention groups in paracyclists or in able-bodied individuals (*p* < 0.05). [Lac] in the warm-up was significantly lower in paracyclists compared to able-bodied individuals (*p* = 0.029). Maximal and average heart rate (HR) as well as PO during the warm-up were all significantly different between the two groups of participants (*p* < 0.05). Data are presented in Table 4.

No significant difference in time to complete the 10 km handcycling TT (Table 5) was found between the three interventions in able-bodied individuals (*p* = 0.61). Furthermore, no significantly different time to complete the TT in paracyclists was found (*p* = 0.80). Paracyclists completed the 10 km TT significantly faster in all three trials compared to able-bodied individuals (*p* < 0.05) and therefore showed a significantly higher PO_average_ compared to able-bodied individuals (*p* < 0.05).

Neither [Lac] (*p* = 0.31) after the time trial nor rated perceived exertion (RPE) (*p* = 0.41) for the trials were significantly different between the three interventions in able-bodied individuals. The same was true in paracyclists between the three interventions (RPE: *p* = 0.43; [Lac]: *p* = 0.68). No significant differences were found in RPE (*p* = 0.08) and [Lac] (*p* = 0.86) concentration between able-bodied individuals and paracyclists. No significant differences between the three interventions were found in HF_average_ in able-bodied individuals (*p* = 0.25) or in paracyclists (*p* = 0.69) during the TT, and there was no difference between the two groups of participants (*p* = 0.11). Data are presented in Table 5.

Out of all 22 participants, only one suffered from gastrointestinal side effects. The participant reported nausea. No significant order effect was found comparing time to complete the first, the second, and the third TT regardless of the supplement in able-bodied individuals (*p* = 0.07) and in paracyclists (*p* = 0.14). PO_average_ was significantly greater in the third TT compared to the second in able-bodied individuals (*p* = 0.039, mean difference + 4.4 W, 95%-CI [0.2; 8.85]) but not in paracycling athletes (*p* = 0.11).

### 3.4. Oxygen Uptake and PO to VO_2_ Ratio

No significant difference was found in VO_2average_ between the three interventions in able-bodied individuals (*p* = 0.15) or in paracyclists (*p* = 0.07). VO_2average_ was found to be significantly higher in paracyclists (VO_2average_: NIT = 35.4 ± 9.5 min/kg, BR = 34.7 ± 10.3 mL/min/kg, PLA = 35.2 ± 9.9 mL/min/kg) compared to able-bodied (VO_2average_: NIT = 29.0 ± 4.2 mL/min/kg, BR = 27.8 ± 4.6 mL/min/kg, PLA = 29.3 ± 4.4 mL/min/kg) individuals (*p* = 0.042). Furthermore, no significant difference was found in PO to VO_2_ ratio between the three interventions in able-bodied individuals (*p* = 0.62) or in paracyclists (*p* = 0.88). PO to VO_2_ ratio was significantly different in some kilometers over the course of the TT in able-bodied individuals (*p* < 0.05). No significant differences in the same parameter were found in the paracyclists (*p* > 0.05) during the TT (Figure 1).

## 4. Discussion

To our knowledge, this is the first study examining the effects of acute nitrate supplementation on 10 km handcycling TT performance. The principal finding of the current investigation was that acute nitrate supplementation did not improve 10 km handcycling TT performance in recreationally active men or in trained paracyclists by BR or by NIT ingestion.

### 4.1. Performance

A single dose of 6 mmol nitrate (BR or NIT) did not improve 10 km handcycling TT performance in able-bodied, upper body trained individuals or in trained paracyclists with a spinal cord injury (Table 5). To date, only a few studies examined the effects of nitrate supplementation on upper body exercise performance [36,37,38,39,43,44]. It is worth mentioning that rowing is a whole body exercise; most of its performance is produced by the lower extremities. Furthermore, these studies focused on short-term exercises, including 4 min maximal effort exercises [37], single [43] or repeated sprint tests [36,38], and 500 m to 2 km TT [36,37,38]. In the study of Peeling et al. [37], the ingestion of 9.6 mmol BR significantly enhanced 500 m kayaking TT performance by 1.7%. However, in the same study, a 4 min TT performance remained unchanged after the intake of 4.8 mmol BR. Comparable results were shown by Hoon et al. [39], where the authors concluded that the ingestion of 8.4 mmol but not 4.2 mmol BR may improve 2 km rowing TT performance [39]. Furthermore, Bond et al. [38] reported improved repeated sprint rowing performance after six days of BR supplementation. The fact that these studies did not reveal any ergogenic effects after the ingestion of a small nitrate dose (e.g., 4.2 to 5 mmol BR) may lead to the assumption that the dose used in the present study (6 mmol BR or NIT) was insufficient to induce any significant effects specifically on upper body exercise performance. Therefore, we hypothesize that a higher dosage of BR or NIT is needed for an acute supplementation protocol to improve 10 km handcycling TT performance. However, of course, a lack of effect of BR supplementation on performance cannot be ruled out.

In the review of Jones [45], it was concluded that nitrate supplementation appears to be more effective in exercises within the duration from 5 to 30 min compared to exercises lasting more than 40 min. Short duration exercises are typically performed at high intensity with predominant fast-twitch muscle fiber contribution and are likely to result in hypoxic and acidic skeletal muscle state. Recalling that the reduction from [NO_3_^−^] to NO is enhanced in hypoxic and acidic conditions [4] and that nitrate has targeted effects on fast-twitch muscle fibers [29,30], it becomes clear that short duration exercises may benefit more from nitrate supplementation. Nevertheless, type II muscle fibers are also recruited during prolonged low-intensity exercise. In the present study, the exercise duration was around 20 min, and the mean intensity was about 85% of VO_2peak_, which indicates that the 10 km handcycling TT would provide optimal conditions for effective nitrate supplementation, particularly in view of the fact that upper body muscles possess a higher contribution of fast-twitch muscle fibers compared to the muscles of the lower body in healthy humans [31,32]. Furthermore, individuals with a spinal cord injury, M. deltoideus, seem to have a higher portion of type I muscle fibers [46]. Thus, nitrate would be expected to be less beneficial in those individuals compared to healthy, able-bodied controls. However, our study showed no improvement in performance in either group of athletes.

To date, only one study examined the effect of chronic nitrate supplementation on whole body exercise performance [38]. They reported an improved repeated 500 m sprint rowing performance after six days of BR supplementation (5.5 mmol per day). The dosage was therefore very similar to the 6 mmol used in our study but, interestingly, the authors revealed beneficial effects on performance. Therefore, it may be suggested that a chronic supplementation protocol may be more beneficial in terms of improving exercise performance compared to an acute supplementation protocol. This assumption is in line with the findings of Vanhatalo et al. [47], who reported that a chronic 15 day BR supplementation (5.2 mmol per day) resulted in a greater effect on incremental cycling exercise performance compared to a single dose of BR [47]. These findings are supported by the two studies of Cermak et al. [15], [48]. In the first study, the authors reported significant improvements in 10 km cycling TT performance after six days of BR supplementation (8 mmol per day) [15]. Conversely, in the second study, acute BR ingestion (8.7 mmol) did not show any positive effects on 1 h cycling performance [48]. The authors assumed that a longer supplementation period (chronic) is necessary to potentiate any ergogenic effects of BR. It is worth noting that, in these two studies, exercise duration and participant characteristics were different; therefore, comparison of the two studies should be taken with caution. A recent investigation by Jo et al. [49] compared the effects of acute (i.e., 2.5 h before exercise) and chronic (i.e., 15 days) nitrate supplementation (8 mmol per day) on 8 km cycling TT performance. The authors reported significant improvements in TT performance after chronic supplementation, whereas acute nitrate ingestion was unable to induce any ergogenic effects. Taking all these findings into account, we assume that the missing ergogenic effect of nitrate supplementation in the present study is partly attributable to the acute supplementation regimen. To date, no study investigated the effect of chronic BR supplementation on upper body exercise performance.

Recent studies reporting no improvement of performance after nitrate supplementation concluded that the fitness level of their participants (VO_2peak_ > 6 min/kg) might play an important role [36,50,51,52,53]. It has been shown that highly trained athletes exhibit increased NO synthase activity (NOS) and a better muscle capillarization compared to untrained individuals [54]. Furthermore, highly trained endurance athletes showed higher plasma nitrite levels [55] and fewer type II muscle fibers compared to recreationally active individuals [56]. All these adaptations to training may result in limited or attenuated effects of nitrate supplementation on performance in trained individuals [45]. In the study of Porcelli et al. [21], nitrate supplementation improved 3 km running TT performance in low to moderate trained participants but not in highly trained athletes (VO_2peak_ > 63 mL/min/kg). Therefore, it was concluded that highly trained athletes are less likely to benefit from nitrate supplementation [21]. In the present study, we tested recreationally active participants (VO_2peak_ = 33.7 ± 4 mL/min/kg) and paracyclists (VO_2peak_ = 38.6 ± 10.5 mL/min/kg). To note, the recorded VO_2peak_ may not reflect the true VO_2max_ value, as during our specific test, only upper body muscles were active. According to the literature [57], we expect the real VO_2peak_ values to be in a range of 45 to 55 mL/min/kg for the able-bodied individuals. Calculating the according value for the paracyclists produces a range between 50 to 60 mL/min/kg, but of course those athletes have secondary conditions such as a limited heart rate [58] or a reduced muscle mass in the core and the upper body [59]. Therefore, we assume that their VO_2peak_ would be even higher, as in able-bodied cyclists. Nevertheless, our able-bodied participants did not fall in the category of highly trained individuals. It seems that the fitness level of those participants would not be a major reason for missing performance enhancing effects.

It is possible that some of our athletes could have been identified as a so-called “responder” with a beneficial effect of either BR or NIT supplementation. Several studies have identified responders and non-responders in their participant samples [51,60,61,62]. The physiological background of these individual responses to nitrate supplementation is presently unknown and requires further investigation. It is suggested that habitual nitrate intake, baseline [NO_2_^−^] concentrations, activity of oral and gut bacteria, training status, and fiber type distribution may play a major role in the appearance of responders and non-responders [45]. Although the present study did not report an ergogenic effect of nitrate supplementation, some participants improved their 10 km TT performance after the ingestion of both BR and NIT. Therefore, we would recommend athletes trialing whether they benefit from nitrate ingestion or not. As long as it is well tolerated, we would not advise athletes against the use of nitrate supplementation. Furthermore, it might possibly be an advantage to follow a chronic supplementation protocol over several days or to use a higher dosage than 6 mmol nitrate.

### 4.2. Oxygen Consumption and PO to VO_2_ Ratio

It is assumed that the ergogenic effect of nitrate supplementation is due to a reduction in VO_2_, as a consequence of enhanced mitochondrial efficiency [23] and/or improved muscle contractile functions [24,25,29]. Larsen et al. [23] demonstrated that following three days of nitrate supplementation VO_2_ at a given workload was significantly reduced by 3% during submaximal cycling exercise. This reduction in VO_2_ was predominately related to the increased mitochondrial P/O ratio (ATP produced by O_2_ consumed) [23]. Further investigations revealed that, after chronic nitrate supplementation, the expression of adenine nucleotide translocase (ANT) was reduced, a protein that is responsible for a substantial part of mitochondrial proton leakage, thus improving mitochondrial efficiency [23]. On the other hand, Bailey et al. [24] demonstrated that the reduction in VO_2_ after six days of BR ingestion was related to a reduced ATP cost of muscle force production (i.e., higher muscle efficiency). However, the exact mechanisms remained unsolved. Furthermore, Hernandez et al. [29] reported that after seven days of nitrate supplementation in mice, force production of fast-twitch muscle fibers was increased. The authors suggested that the result was due to an increased expression of muscle Ca^2+^-handling proteins in fast-twitch muscle fibers after nitrate supplementation [28]. In addition to these findings, Haider et al. [25] revealed that seven days of BR ingestion enhanced excitation–contraction coupling in human skeletal muscles. Thus, upper body muscles in general would benefit more, as they tend to have more type II muscle fibers [32]. However, a lesion of the spinal cord (and therefore a predominant use of upper body muscle) seems to lead to an increase in muscle type I fibers [46]. Therefore, we expected an increase in performance following nitrate ingestion in able-bodied individuals but possibly not in paracyclists with a spinal cord injury.

To date, it was shown that dietary nitrate supplementation either by BR or NIT reduces steady-state VO_2_ (3 to 14%) for a given PO during submaximal exercises irrespective of exercise modality (cycling, running, kayaking, etc.) [20,47,63,64]. However, not all studies were able to detect such a reduction of VO_2_ [65,66]. Our study (Figure 1) showed that PO to VO_2_ ratio tended to be higher in the BR trial compared to NIT and PLA in able-bodied individuals with a significant difference in some km during the TT. However, VO_2_ alone was not significantly lower in the BR trial. To summarize the most recent literature, it was noticed that the PO/VO_2_ ratio increased to a larger extent during cycling (i.e., lower body: 7 to 11%) compared to handcycling and kayaking (i.e., upper body: 2 to 3%) [16,37]. It could be hypothesized that the smaller increase in the PO/VO_2_ ratio is simply a consequence of a smaller active muscle mass during upper body exercise compared to cycling. On the other hand, upper body exercises rely largely on fast-twitch muscle fibers [67,68], and therefore it would be expected that upper body exercises may show greater benefits compared to cycling or running. However, the results of the present study do not support this assumption.

In the present study, we found an increase in PO/VO_2_ ratio in some km of the TT after BR ingestion but not after NIT despite the same dosages (6 mmol) of the supplements (Figure 1a). This observation is contradictory, since several studies reported a reduced VO_2_ during exercise following NIT supplementation [23,63]. In contrast to our study, these investigations used a chronic supplementation protocol. Thus far, few studies examined the effects of acute NIT ingestion on VO_2_ [65,66]. Bescos et al. [66] reported that 10 mg/kg BM nitrate (~11 mmol) did not reduce VO_2_ during submaximal cycling in trained endurance athletes. These results are in close agreement with those obtained by Flueck et al. [65], where different dosages of NIT (3, 6, and 12 mmol) did not reduce VO_2_ response of moderate- and severe-intensity cycling. On the other hand, BR supplementation appeared to elicit greater effects on VO_2_ due to the additional antioxidants, which facilitate nitrate reduction. Additionally, BR contains polyphenols with a potential for ergogenic effects in performance [69]. It is possible that BR might be more beneficial in reducing VO_2_ compared to NIT, but more research is needed to confirm those assumptions.

### 4.3. Other Parameters

In the present study, neither BR nor NIT had any significant influence on systolic or diastolic blood pressure at rest (Table 2). Similar findings were shown in a similar study [65], whereas only the ingestion of a higher BR dosage seemed to result in a significantly lower blood pressure at rest. It seems that a dose-dependent decrease in blood pressure occurs after the ingestion of nitrate supplementation [10]. The mechanism by which dietary nitrate supplementation lowers blood pressure may be explained by the increased bioavailability of NO via the nitrate-nitrite-NO-pathway [70,71]. NO is known as a potent vasodilator that governs systemic blood pressure by reducing arterial pressure and peripheral resistance [4]. Therefore, a sufficient amount of nitrate needs to be ingested to increase the concentration of NO, thus inducing a reduction in blood pressure. Previous studies concluded that, compared to nitrate salts, BR has a greater potential to reduce systolic blood pressure due to the additional antioxidants in BR that facilitate the reduction of nitrite to NO [65]. However, the findings of the present study did not confirm this assumption.

Resting levels of [NO_3_^−^] seemed to be lower in the paracyclists (~38 uM) compared to able-bodied individuals (~66 uM). To our knowledge, no other study measured resting [NO_3_^−^] concentration in participants with a spinal cord injury. Therefore, we might only speculate about the reason for such a difference. Possible reasons might be the lower energy or the nitrate intake in general due to lower active muscle mass resulting in lower energy expenditure. Another explanation might involve the slower gastrointestinal transition time in those participants due to the impaired nervous system [72]. This might also affect the absorption of nutrients. Significant increases in [NO_2_^−^] and [NO_3_^−^] concentrations after NIT and BR were found in both groups of athletes. Interestingly, able-bodied individuals showed significantly higher [NO_2_^−^] concentrations after BR consumption compared to paracyclists, whereas paracyclists showed significantly higher [NO_3_^−^] concentrations after BR and NIT ingestion compared to able-bodied individuals. It seems possible that reduction from [NO_3_^−^] to [NO_2_^−^] was reduced in the paracyclists and therefore a higher [NO_3_^−^] was present 3 h after the ingestion. Such a hypothesis needs to be confirmed in future studies. No significant differences were found in [NO_2_^−^] and [NO_3_^−^] concentrations between NIT and BR ingestion in the same group. These findings seemed to be very similar to the previous study with a 6 mmol NIT and BR intervention in able-bodied individuals [65]. Thus, the results for [NO_2_^−^] and [NO_3_^−^] concentrations in this study seem to be consistent with the literature.

### 4.4. Limitations

One major limitation of this study was that we did not use a nitrate-depleted BR as a placebo supplement. The participants were possibly able to distinguish between BR and NIT or between BR and PLA. Therefore, we cannot exclude the possibility that our findings were influenced by a potential placebo effect in the BR group. However, the participants were not able to distinguish between PLA and NIT because of similar taste, smell, and appearance.

Additionally, there was a significant difference between the two groups. Able-bodied individuals showed a significantly higher body mass compared to paracyclists. This fact evolved mainly from the loss of muscle mass in the legs of athletes with a spinal cord injury due to immobilization. Furthermore, the age was significantly different between the two groups, as paracyclists were older. This is a known phenomenon in Paralympic sports, as most athletes start their career after the incident of the injury. Last but not least, both groups showed a different fitness level, as paracyclists appeared to be more highly trained than able-bodied individuals. All these issues might have influenced the outcome of the study and therefore needed to be acknowledged.

Finally, although our able-bodied participants were used to upper body exercise, we still reported a significant order effect on 10 km handcycling TT performance in this group of participants. Therefore, we assume that only one familiarization trial was insufficient in the present study. We cannot exclude the possibility that the effects of dietary nitrate on exercise performance and VO_2_ in the present study were masked or blunted by the observed order effect.

## 5. Conclusions

Our study demonstrated that acute nitrate supplementation either by BR or NIT (6 mmol) did not improve 10 km handcycling TT performance in recreationally active men or in paracyclists. BR ingestion seemed to increase PO to VO_2_ ratio in some km of the TT compared to PLA in able-bodied individuals but not in paracyclists. Overall, the present results do not support previous findings where acute nitrate supplementation enhanced short-term, high-intensity TT performance [36,37,38] in upper body exercise. Further research is required to examine optimal supplementation strategies and effects of nitrate supplementation on upper body exercise performance, especially in wheelchair athletes with physiological adaptations following the incident of the spinal cord injury.

## Figures and Tables

**Figure 1 nutrients-11-01642-f001:**
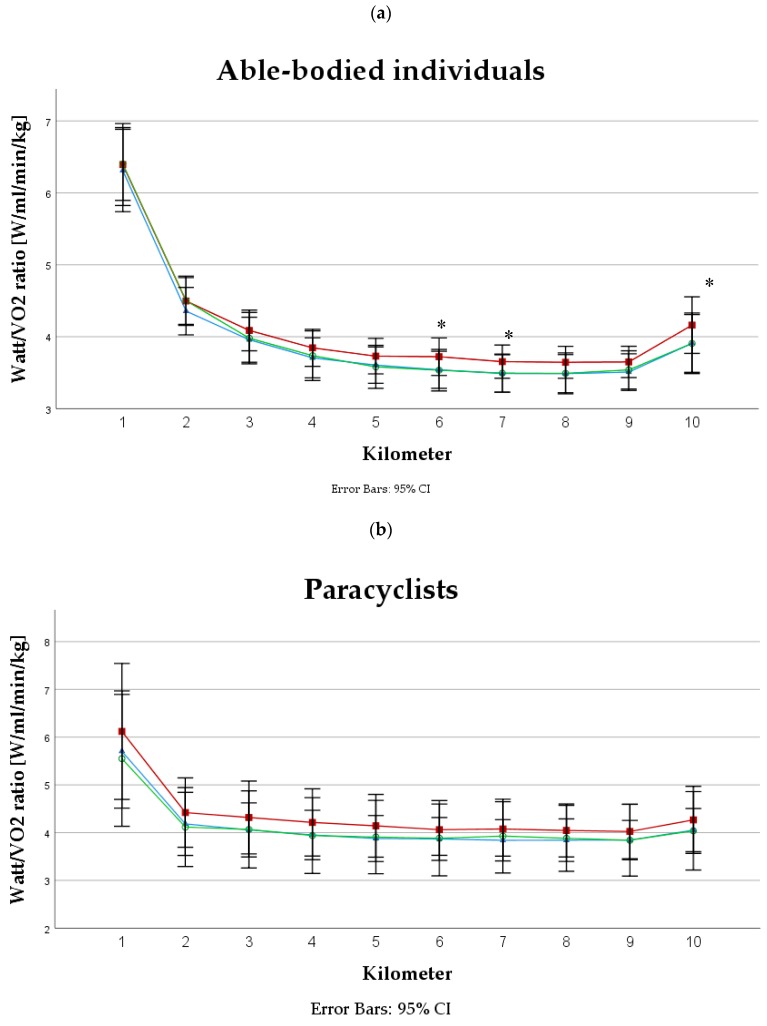
PO to VO_2_ ratio in (**a**) able-bodied individuals and (**b**) in paracyclists during a 10 km handcycling TT. Note: * = significant difference (*p* < 0.05), red squares (■) = beetroot juice intervention, blue triangles (▲) = sodium nitrate intervention and green circles (○) = placebo intervention.

**Table 1 nutrients-11-01642-t001:** Characteristics of the eight paracyclists.

ID	Age [year]	Height [cm]	Body Mass [kg]	VO_2peak_ [ml/min/kg]	PO_max_ [W]	Lesion Level	Category
**1**	42	188	66	41.5	200	Th5	MH3
**2**	34	169	72	29.7	144	L4	MH4
**3**	29	170	47	42.8	151	C4	MH2
**4**	54	178	75	49.7	222	Th12	MH5
**5**	32	173	60	36.9	180	Th3	MH3
**6**	61	178	61	45.9	208	Th4	MH3
**7**	40	165	64	44.8	220	Th10	MH4
**8**	35	190	76	17.3	98	C6	MH1
	40 ± 11	176 ± 9	65 ± 9	38.6 ± 10.5	178 ± 44		

Note: Data presented as mean ± standard deviation, PO_max_ = maximal power output in the ramp test, VO_2peak_ = peak oxygen uptake measured during the ramp test.

**Table 2 nutrients-11-01642-t002:** Blood pressure and heart rate before and after ingestion of the supplements.

	Able-Bodied Individuals			Paracyclists		
	PLA	NIT	BR	PLA	NIT	BR
**Systolic BP [mmHg]**						
Pre	124 ± 9	125 ± 9	124 ± 10	129 ± 16	130 ± 19	128 ± 17
Post	125 ± 9	125 ± 13	125 ± 10	134 ± 14	131 ± 18	135 ± 17
**Diastolic BP [mmHg]**						
Pre	77 ± 7	78 ± 10	79 ± 9	79 ± 15	78 ± 18	79 ± 15
Post	80 ± 8	80 ± 9	82 ± 9	84 ± 13	83 ± 16	84 ± 14
**HR [bpm]**						
Pre	61 ± 6	62 ± 6	61 ±7	70 ± 17	70 ± 14	67 ± 17
Post	56 ± 7	55 ± 7	55 ± 9	61 ± 10	62 ± 10	59 ± 12

Note: data presented as mean ± SD, HR = heart rate, BP = blood pressure, PLA = placebo, NIT = sodium nitrate, BR = beetroot juice, bpm = beats per minute.

**Table 3 nutrients-11-01642-t003:** Plasma [NO_3_^−^] and [NO_2_^−^] in able-bodied individuals and paracycling pre and post ingestion of BR, NIT, and PLA.

		Able-Bodied Individuals		Paracyclists	
		Pre	Post	Pre	Post
**Plasma [NO_3_^−^]** in uM	PLA	68.5 ± 8.4	67.9 ± 8.6	37.9 ± 18.8	38.1 ±18.9
	NIT	64.8 ±10.9	278.7 ± 154.9 *^†^	36.8 ± 19.9	85.9 ± 73.2 *^†^
	BR	63.5 ± 8.9	273.7 ± 82.8 *^†^	38.8 ± 17.9	125.8 ± 99.3 *^†^
**Plasma [NO_2_^−^]** in nM	PLA	44.7 ± 21.5	41.2 ± 28.7	66.9 ± 27.9	92.3 ± 109.9
	NIT	56.7 ± 22.6	141.2 ± 75.7 *^†^	57.7 ± 11.6	136.7 ± 69.6 *^†^
	BR	61.5 ± 53.8	121.2 ± 57.3 *^†^	108.7 ± 134.3 ^†^	263.3 ± 159.2 *^†^

Note: PLA = placebo, NIT = sodium nitrate, BR = beetroot juice, * significant difference (*p* < 0.05) compared to pre ingestion, ^†^ = significant difference (*p* < 0.05) to PLA, pre = before ingestion of the supplement, post = 3 h after ingestion of the supplement.

**Table 4 nutrients-11-01642-t004:** Warm-up data for the two groups.

	Able-Bodied Individuals			Paracyclists		
	PLA	NIT	BR	PLA	NIT	BR
PO_average_ [W]	55 ± 19	58 ± 16	55 ± 18	90 ± 31 *	95 ± 31 *	88 ± 29 *
HR_average_ [bpm]	93 ± 19	95 ± 15	94 ± 16	119 ± 20 *	115 ± 18 *	111 ± 19 *
HR_max_ [bpm]	105 ± 21	107 ± 15	107 ± 16	140 ± 25 *	137 ± 23 *	137 ± 26 *
RPE [6; 20]	11 [7; 13]	11 [7; 12]	11 [7; 13]	12 [8; 14]	11 [7; 12]	12 [8; 13]
[Lac] [mmol/L]	2.57 ± 1.43	2.67 ± 1.34	2.68 ± 1.13	1.89 ± 0.86 *	1.74 ± 0.77 *	1.87 ± 0.73 *

Note: Data presented as mean ± standard deviation and mean [min; max] for RPE, PLA = placebo, NIT = sodium nitrate, BR = beetroot juice, [Lac] = plasma lactate concentration after the time trial, RPE = rated perceived exertion, HR = heart rate, PO = power output, * = significant difference (*p* < 0.05) compared to able-bodied individuals.

**Table 5 nutrients-11-01642-t005:** Data from the 10 km time trail (TT) performance in both groups of participants.

	Able-Bodied Individuals			Paracyclists		
	PLA	NIT	BR	PLA	NIT	BR
Time to complete [s]	1182 ± 84	1194 ± 102	1195 ± 94	1106 ± 247 *	1091 ± 235 *	1071 ± 199 *
PO_average_ [W]	114 ± 16	113 ± 20	112 ± 17	142 ± 49	144 ± 44	145 ± 42
HR_average_ [bpm]	152 ± 13	153 ± 9	150 ± 11	162 ± 27 *	158± 30 *	158 ± 31 *
HR_max_ [bpm]	174 ± 10	175± 8	175 ± 9	176 ± 24	170 ± 31	173 ± 28
RPE [6; 20]	18 [16; 20]	19 [17; 20]	18.5 [17; 20]	19 [17; 20]	19 [17; 20]	19 [17; 20]
[Lac] [mmol/L]	11.94 ± 2.40	12.28 ± 2.84	11.66 ± 2.15	10.13 ± 5.98	10.05 ± 6.37	9.92 ± 5.99

Note: Data presented as mean ± standard deviation and mean [min; max] for RPE, RPE = rated perceived exertion, HR_average_ = average heart rate during the test, HR_max_ = maximal heart rate during the test, PO_average_ = average power output during the test, [Lac] = lactate concentration after the test, NIT = sodium nitrate, BR = beetroot juice, PLA = placebo, * = significant (*p* < 0.05) difference between the two groups of participants.

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
