# Peer review of "Influence of Equimolar Doses of Beetroot Juice and Sodium Nitrate on Time Trial Performance in Handcycling"

_nutrients, 2019, doi:10.3390/nu11071642_

Round 1
Reviewer 1 Report
Overall, this is an interesting study, adding new knowledge to the field. The main novelty resides on the type of exercise studied, together with the inclusion of para-cyclists.
Beetroot has no molecular weight and therefore cannot have a molar calculation, this need to be clarified in the abstract (see also Line 70: this is explained in line 124)
Line 54. No single study in humans has demonstrated that BR juice increases muscle blood flow and O2 delivery in humans. Ref 25 refers to a study in rodents. Also, previous studies using an intra-arterial infusion of vasodilators in humans demonstrated that this type of intervention does not improve performance or peak leg blood flow during whole-body exercise (see PMID: 17932136; PMID: 16914431). Thus, it is essential than in the introduction, and throughout the manuscript, you differentiate between findings obtained in animal models from those obtained in humans.
L58 Reference to PMID: 26589128 is also relevant to this point.
L59 Reference 31 is a review from 1986. Later studies using a-v differences and stable isotopes demonstrated that the arms rely more on glycolytic metabolism and produce more lactate (see REF: PMID: 12388120).
L79, What is the meaning of “3 units per week”?
L99. It is said that TTs were at least two days apart. This is too short; and, in Lines 281, you report a potential carry-over effect. This fact does not affect the interpretation of the results obtained as far as treatments were distributed in balance day in trial 1, 2 and 3. Was this the case?
L109. In what position and how was blood pressure and HR measured.
L128. The supplier of the Nitrate powder and the reference of the product should be reported.
L131. Was any question asked to the subjects to ascertain whether the concealment was efficient? i.e., how many subjects were able to guess what were they drinking before the TT?
Line 214. Please, show the data in a Table. This is relevant given that previous papers reported reduced blood pressure after BR juice.
Line 293-294. How was the statistical analysis done? This type of comparison requires statistically significant main o interaction effects in the ANOVA for repeated measures analysis. Was this the case? How did you correct for multiple comparisons? I think this finding is irrelevant.
Line 330. The alternative hypothesis, i.e., lack of effect of BR supplementation on performance cannot be ruled out. Some studies report effects while others fail to find differences. This has occurred with other “ergogenic aids” in the past, initial studies reporting clear results, and then the effect vanishes over time.
Line 317. I agree you should not consider rowing as an upper body exercise it pertains to the whole-body exercise category. Please, correct. Besides, in your case, the upper-body exercise done by the paracyclists is a “real” upper body exercise (this support the novelty of your study).
L337. Nevertheless, with fatigue type II fibers are also recruited during prolonged low-intensity exercise.
L346. Rowing is not upper-body exercise.
L407-408, P/O ratio is the other way around, ATP produced by O2 consumed. Higher P/O ratio, the better the efficiency (good for performance).
L424-426. Several studies agree on this, but not all (see your lines 443-444).
L426-28. You cannot make this statement, which is against what the statistical analysis reflects.
L428-29. There is nothing wrong about reporting your results (they are good). Do not try to align your results with previous studies, when your date does not confirm previous studies. There no difference, so you cannot claim about a clinically relevant difference (in addition, there is nothing “clinical” in this study).
L437-438. This is an overstatement given the results of the statistical analysis applied.
L448. Beetroot juice must contain polyphenols, which are not present in a sodium nitrate aqueous solution (similar to your lines 460-61). Several polyphenols have ergogenic effects…
L484-85. I do not agree with the conclusion that “BR ingestion resulted in a significantly increased PO to VO2 ratio compared to PLA during the TT in able-bodied individuals.” The mean Power/VO2 ratio of total TT was not statistically different, and therefore, post-hoc comparisons are not permitted without a significant main effect or interaction.
Figure 1. This information can be better interpreted if presented in a Table.
Figure 2. It is not possible to distinguish between conditions; please use colour symbols contrasting more shapely.
Author Response
Beetroot has no molecular weight and therefore cannot have a molar calculation, this need to be clarified in the abstract (see also Line 70: this is explained in line 124)
Thank you for this comment. Of course, there is no molecular weight for BR. This issue should be clarified now.
Line 54. No single study in humans has demonstrated that BR juice increases muscle blood flow and O2 delivery in humans. Ref 25 refers to a study in rodents. Also, previous studies using an intra-arterial infusion of vasodilators in humans demonstrated that this type of intervention does not improve performance or peak leg blood flow during whole-body exercise (see PMID: 17932136; PMID: 16914431). Thus, it is essential than in the introduction, and throughout the manuscript, you differentiate between findings obtained in animal models from those obtained in humans.
Thank you for this comment. This is a very important point and has been changed in the introduction.
L58 Reference to PMID: 26589128 is also relevant to this point.
Thank you for this suggestion. The reference has been added.
L59 Reference 31 is a review from 1986. Later studies using a-v differences and stable isotopes demonstrated that the arms rely more on glycolytic metabolism and produce more lactate (see REF: PMID: 12388120).
The paragraph was changed following your suggestion.
L79, What is the meaning of “3 units per week”?
We clarified this issue by stating “sessions per week”.
L99. It is said that TTs were at least two days apart. This is too short; and, in Lines 281, you report a potential carry-over effect. This fact does not affect the interpretation of the results obtained as far as treatments were distributed in balance day in trial 1, 2 and 3. Was this the case?
We clarified the issue of how much time has been between the two trials. It has been at least 48 h. In term of washout period, this should have been enough as the results for the blood samples have shown.
The order effect is described in the results sections as following: “. No significant order effect was found comparing time to complete the first, second and third TT regardless of the supplement in able-bodied individuals (p=0.07) and in paracyclists (p=0.14). POaverage was significantly greater in the third TT compared to the second in able-bodied individuals (p=0.039, mean difference + 4.4 W, 95%-CI [0.2; 8.85]) but not in paracycling athletes (p=0.11).” This means that time to complete the TT was not significantly influenced by a order effect. Able-bodied individuals seemed to have a higher power output in the third trial compared to the second. Therefore, we might expect no shortage of recovery time even more a learning effect of pacing accordingly (as handcycling was not their preferred event of upper body exercise (able-bodied individuals)). A brief discussion of this is mentioned in the limitations section. Hopefully, we could provide you a satisfying answer to your question.
L109. In what position and how was blood pressure and HR measured.
Thank you for this comment. We added that the forearm was placed at heart level. “resting blood pressure and heart rate were measured in triplets in a sitting position at the wrist using an automated cuff (Boso medistars, Bosch + Sohn GmbH, Jungingen, Germany). The forearm was placed at heart level.”
L128. The supplier of the Nitrate powder and the reference of the product should be reported.
This was added to the manuscript.
L131. Was any question asked to the subjects to ascertain whether the concealment was efficient? i.e., how many subjects were able to guess what were they drinking before the TT?
“Shortly after each exercise test, participants were asked about the supplement they received.” This information is in the manuscript. We have added the other information concerning the right guess after the test.
Line 214. Please, show the data in a Table. This is relevant given that previous papers reported reduced blood pressure after BR juice.
Thank you for this suggestion. Data has been added as a table.
Line 293-294. How was the statistical analysis done? This type of comparison requires statistically significant main o interaction effects in the ANOVA for repeated measures analysis. Was this the case? How did you correct for multiple comparisons? I think this finding is irrelevant.
Thank you for this comment. The statistics were performed as following: “The PO and PO/VO2 ratio data were graphed for the 10-km distance and analyzed using a two-way (supplement x distance) repeated-measures ANOVA with Bonferroni correction. In case of violation of sphericity, Greenhouse-Geisser correction was applied. The effect of order on TT performance was analyzed using a one-way repeated-measures ANOVA and pairwise t-tests with Bonferroni correction.” This is described in the methods section.
Line 330. The alternative hypothesis, i.e., lack of effect of BR supplementation on performance cannot be ruled out. Some studies report effects while others fail to find differences. This has occurred with other “ergogenic aids” in the past, initial studies reporting clear results, and then the effect vanishes over time.
We are grateful for this comment. The sentence was added.
Line 317. I agree you should not consider rowing as an upper body exercise it pertains to the whole-body exercise category. Please, correct. Besides, in your case, the upper-body exercise done by the paracyclists is a “real” upper body exercise (this support the novelty of your study).
Thank you for this comment, the paragraph was changed accordingly.
L337. Nevertheless, with fatigue type II fibers are also recruited during prolonged low-intensity exercise.
This was added to the manuscript.
L346. Rowing is not upper-body exercise.
Thank you, this was changed.
L407-408, P/O ratio is the other way around, ATP produced by O2 consumed. Higher P/O ratio, the better the efficiency (good for performance).
This was changed according your suggestion.
L424-426. Several studies agree on this, but not all (see your lines 443-444).
This paragraph has been changed.
L426-28. You cannot make this statement, which is against what the statistical analysis reflects.
Thank you for this comment. This paragraph has been changed.
L428-29. There is nothing wrong about reporting your results (they are good). Do not try to align your results with previous studies, when your date does not confirm previous studies. There no difference, so you cannot claim about a clinically relevant difference (in addition, there is nothing “clinical” in this study).
Thank you for this comment. This paragraph has been changed.
L437-438. This is an overstatement given the results of the statistical analysis applied.
This has been changed.
L448. Beetroot juice must contain polyphenols, which are not present in a sodium nitrate aqueous solution (similar to your lines 460-61). Several polyphenols have ergogenic effects…
We are grateful for this comment and added a sentence and some literature.
L484-85. I do not agree with the conclusion that “BR ingestion resulted in a significantly increased PO to VO2 ratio compared to PLA during the TT in able-bodied individuals.” The mean Power/VO2 ratio of total TT was not statistically different, and therefore, post-hoc comparisons are not permitted without a significant main effect or interaction.
Thank you for this comment, this was changed accordingly. The mean of PO/VO2 ratio over the whole 10km was not significantly different but there was an effect comparing the different km of the 10km with the intervention.
Figure 1. This information can be better interpreted if presented in a Table.
Figure 1 was deleted and the information from the Figure was added to Table 5.
Figure 2. It is not possible to distinguish between conditions; please use colour symbols contrasting more shapely.
We are thankful for this comment. The Figure was changed according your suggestion to use color symbols. Nevertheless, it seems difficult to distinguish between the three intervention. But, this highlights, that there were no major differences only in km 6, 7 and 10 in AB.
Reviewer 2 Report
Comments on manuscript “Influence of equimolar doses of beetroot juice and sodium nitrate on time trial performance in handcycling” by Flueck et al. in Nutrients journal.
Authors compare the effect of sodium nitrate, beetroot juice and placebo supplementation on time to complete 10km distance using handcycling device. Their participants are either healthy volunteers or paracyclist. The major described finding is that dietary supplementation by 85ml of 6mM nitrate in either form did not led to significant faster completion of 10km distance by either group. The novelty of the paper is that they are using true upper body exercise in order to separate body parts with predominantly fast twitch muscle, which are, supposedly, more sensitive and responsive to nitrate supplementation.
I have few comments and corrections:
1. the reference 3 mentioned in the Introduction on line 34 does not specifically say that there is a temporary increase of NO in muscle, it only implies it from increase in plasma and, therefore one could expect it in the muscle (or any other organ for that matter), derived from the blood flow. Currently, the only human study directly showing the increased nitrate in the muscle tissue is Nyakairu et al. L. Appl. Physiol (1985) 2017, 23:637-644.
2. There is an error in reporting NIT supplementation on page3, line 127: “For the preparation of the 6 mmol NIT supplements, the amount of 372 g of sodium nitrate was weighted and dissolved in 85 ml plain water”. 6mmol of NaNO3 (MW 85) is 510 mg, not 372g.
3. In the same paragraph, line 130 – 0.5 g of sodium nitrate in a shot of 85ml will have a slightly salty taste, it would’ve been better to use NaCl as a placebo to dissimulate the taste better – just a comment.
4. On page 4, line 180 – I understand that NIT refers to sodium nitrate, but it could be spelled out for more clarity.
5. When comparing the two groups there are some significant differences that could influence results: body weight of healthy volunteers is 82±9 kg vs. 65±9kg for paracyclists, which is a difference of almost 25%. Therefore, the dose of nitrate per kg body weight is significantly different. Also, age difference is significant - 28±7years vs 40±11years – paracyclists are older and, much better trained that the healthy volunteers. I understand that in spite of these differences there was no effect observed, but I believe it is important to acknowledge the differences between two groups.
6. A very intriguing results are presented in Table 2. I believe some of them should be discussed in more details in discussion:
- When comparing plasma nitrate between healthy volunteers and paracyclists at pre-consumption level, there is a significantly higher level of nitrate in healthy volunteers vs paracyclists – 66uM vs 38uM. Is there any explanation for this?
- As noticed in manuscript, both nitrate and beets increased plasma nitrate in both groups, which is a trivial result. However, there is a significant qualitative difference between healthy volunteer and paracyclist’s response to increased dietary nitrate: both, nitrate and beets increased the levels of nitrate to the same level in healthy volunteers, but in paracyclists, beets were much more effective than nitrate itself. Could some more discussion of this later fact be added? There is only a short description of the fact at the end of the discussion (lines 463-469) but no explanation.
7. Still in table 2, now moving to nitrite.
- In general, pre-consumption levels of plasma nitrite in both groups are extremely low. In general, most groups working on nitrite/nitrate agree that plasma nitrite is usually between 200nM up to 1 uM. In the present table values are between 45 – 109nM, which is significantly lower than expected. Is there any known reason for this?
- And again, as in the case of nitrate, we have the same rend between two groups – beets tend to be better nitrate supply for paracyclists than for healthy volunteers. What could be the possible cause? I could not find a discussion about this anywhere.
As far as the rest of the paper goes, I believe the study was carefully designed and executes and as such it adds into the sum of the studies about the effect of nitrate in one form or the other on the exercise performance. The important novelty and merit of this study is in using exclusively upper body exercise, which had not been done so far. Since there is a hypothesis that nitrate should be more effective on fast twitch fibers, it is important to perform this kind of study, even if the results don’t support the hypothesis.
I my opinion, the points 6 and 7 above make this study stand apart from the rest of similar studies in exercise field, however, the explanation need to be better developed in the discussion.
Author Response
Authors compare the effect of sodium nitrate, beetroot juice and placebo supplementation on time to complete 10km distance using handcycling device. Their participants are either healthy volunteers or paracyclist. The major described finding is that dietary supplementation by 85ml of 6mM nitrate in either form did not led to significant faster completion of 10km distance by either group. The novelty of the paper is that they are using true upper body exercise in order to separate body parts with predominantly fast twitch muscle, which are, supposedly, more sensitive and responsive to nitrate supplementation.
I have few comments and corrections:
1. the reference 3 mentioned in the Introduction on line 34 does not specifically say that there is a temporary increase of NO in muscle, it only implies it from increase in plasma and, therefore one could expect it in the muscle (or any other organ for that matter), derived from the blood flow. Currently, the only human study directly showing the increased nitrate in the muscle tissue is Nyakairu et al. L. Appl. Physiol (1985) 2017, 23:637-644.
We are very grateful for this comment and changed it accordingly.
2. There is an error in reporting NIT supplementation on page3, line 127: “For the preparation of the 6 mmol NIT supplements, the amount of 372 g of sodium nitrate was weighted and dissolved in 85 ml plain water”. 6mmol of NaNO3 (MW 85) is 510 mg, not 372g.
Thank you for this comment, this was changed accordingly.
3. In the same paragraph, line 130 – 0.5 g of sodium nitrate in a shot of 85ml will have a slightly salty taste, it would’ve been better to use NaCl as a placebo to dissimulate the taste better – just a comment.
We agree with your comment. This has to be done in a future study. Nevertheless, only a few participants were able to distinguish between placebo and NIT.
4. On page 4, line 180 – I understand that NIT refers to sodium nitrate, but it could be spelled out for more clarity.
Yes, of course, this must be changed.
5. When comparing the two groups there are some significant differences that could influence results: body weight of healthy volunteers is 82±9 kg vs. 65±9kg for paracyclists, which is a difference of almost 25%. Therefore, the dose of nitrate per kg body weight is significantly different. Also, age difference is significant - 28±7years vs 40±11years – paracyclists are older and, much better trained that the healthy volunteers. I understand that in spite of these differences there was no effect observed, but I believe it is important to acknowledge the differences between two groups.
We are grateful for this comment. A section in the limitations was added to acknowledge those issues. Those are very important concerns. Nevertheless, those differences are known in Paralympic sports and therefore, it seems not realistic to match groups perfectly in future studies.
6. A very intriguing results are presented in Table 2. I believe some of them should be discussed in more details in discussion:
- When comparing plasma nitrate between healthy volunteers and paracyclists at pre-consumption level, there is a significantly higher level of nitrate in healthy volunteers vs paracyclists – 66uM vs 38uM. Is there any explanation for this?
Thank you for this comment on nitrate levels pre ingestion. We can only speculate on possible explanations. One reason might involve the lower energy intake in the paracyclists in general due to a lower energy expenditure. Thus, also the intake of dietary nitrates in general is possibly lower. Additionally, their sympathetic nervous system is impaired due to the lesion and therefore, gastrointestinal transition time prolonged. This again might affect absorption rate of nutrients. A brief discussion of those speculations is added to the manuscript.
- As noticed in manuscript, both nitrate and beets increased plasma nitrate in both groups, which is a trivial result. However, there is a significant qualitative difference between healthy volunteer and paracyclist’s response to increased dietary nitrate: both, nitrate and beets increased the levels of nitrate to the same level in healthy volunteers, but in paracyclists, beets were much more effective than nitrate itself. Could some more discussion of this later fact be added? There is only a short description of the fact at the end of the discussion (lines 463-469) but no explanation.
There was no significant difference between post nitrate levels between NIT and BR in paracyclists. Possibly, the BR levels were higher due to a higher pre ingestion level in the BR trial for NO2-. Another explanation for this high mean is, that one athlete had a value around 600 nM.
7. Still in table 2, now moving to nitrite.
- In general, pre-consumption levels of plasma nitrite in both groups are extremely low. In general, most groups working on nitrite/nitrate agree that plasma nitrite is usually between 200nM up to 1 uM. In the present table values are between 45 – 109nM, which is significantly lower than expected. Is there any known reason for this?
Thank you for this comment. We observed in a previous study very similar values of resting plasma nitrite levels (Flueck et al. 2016). Comparing the values to Wylie et al. 2013 where they studied the pharmacokinetic response of different beetroot juice dosages, they showed also very similar resting nitrite concentrations (Wylie et al. 2013). Therefore, we did not include any discussion into the manuscript.
- And again, as in the case of nitrate, we have the same rend between two groups – beets tend to be better nitrate supply for paracyclists than for healthy volunteers. What could be the possible cause? I could not find a discussion about this anywhere.
Again, no significant difference between NIT and BR for the same group was found. The higher mean value for BR might evolve from one athlete with a high plasma nitrite and nitrate level after BR consumption. We don`t think, that there is a need for a discussion as the results didn`t show any significant difference.
As far as the rest of the paper goes, I believe the study was carefully designed and executes and as such it adds into the sum of the studies about the effect of nitrate in one form or the other on the exercise performance. The important novelty and merit of this study is in using exclusively upper body exercise, which had not been done so far. Since there is a hypothesis that nitrate should be more effective on fast twitch fibers, it is important to perform this kind of study, even if the results don’t support the hypothesis.